# Evaluating the Implementation and Effectiveness of the SWITCH–MS: An Ecological, Multi-Component Adolescent Obesity Prevention Intervention

**DOI:** 10.3390/ijerph17155401

**Published:** 2020-07-27

**Authors:** Senlin Chen, Richard R. Rosenkranz, Gabriella M. McLoughlin, Spyridoula Vazou, Lorraine Lanningham-Foster, Douglas A. Gentile, David A. Dzewaltowski

**Affiliations:** 1School of Kinesiology, Louisiana State University, Baton Rouge, LA 70803, USA; 2Department of Food, Nutrition, Dietetics, & Health, Kansas State University, Manhattan, KS 66506, USA; ricardo@ksu.edu; 3College of Human Sciences and Education, Iowa State University, Ames, IA 50011, USA; gmclough@iastate.edu (G.M.M.); svazou@iastate.edu (S.V.); lmlf@iastate.edu (L.L.-F.); dgentile@iastate.edu (D.A.G.); 4University of Nebraska Medical Center, Omaha, NE 69198, USA; david.dzewaltowski@unmc.edu

**Keywords:** healthy-living behaviors, implementation science, obesity prevention, program evaluation, school wellness

## Abstract

*Background*: The purpose of this study was to evaluate the implementation and effectiveness of an ecological, multi-component adolescent obesity prevention intervention called School Wellness Integration Targeting Child Health–Middle School (SWITCH–MS). *Methods*: Following the effectiveness-implementation hybrid type 3 quasi-experimental design, seven middle schools (377 students) in Iowa, United States, were stratified into “experienced” (*n* = 3; 110 students) or “inexperienced” (*n* = 4; 267 students) groups to receive the 12-week SWITCH–MS intervention. To evaluate implementation, school informants (*n* = 10) responded to a survey and students completed behavioral tracking in the classroom on a website. For effectiveness evaluation, students in 6th, 7th, and 8th grades completed a validated questionnaire before and after intervention, to measure behaviors of physical activity (PA; “Do”), screen-based activity (“View”), and fruits and vegetable consumption (“Chew”). *Results*: The two groups of schools showed similar levels of implementation for best practices, awareness, and engagement. Behavioral tracking rate favored the experienced schools early on (47.5% vs. 11.7%), but differences leveled off in weeks 3–12 (sustained at 30.1–44.3%). Linear mixed models demonstrated significant time effects for “Do” (at school and out of school; *p* < 0.01) and “View” behaviors (*p* = 0.02), after controlling for student- and school-level covariates. *Conclusions*: This study demonstrates that prior experience with SWITCH–MS may not be a prominent factor for implementation and effectiveness, although greater experience is associated with favorable behavioral tracking when the intervention is first launched.

## 1. Introduction

Childhood obesity remains a major global public health concern with over 340 million children and adolescents being classified overweight or obese in 2016 [1]. In the United States, more than one in five adolescents is obese [2], and obesity prevalence in rural areas is higher than that in suburban areas [3]. More youth obesity prevention interventions that allow for adaptations in implementation in rural areas are warranted to reduce health disparity [3]. Weight change is determined by the balance between caloric intake and expenditure [4]. Interventions that work to influence healthy-living behaviors such as physical activity, diet, and screen-time-based sedentary behaviors across environmental settings have shown effectiveness to prevent and control obesity in youth [5,6]. These interventions often entail ecological, multi-level, and multi-component strategies across environmental settings such as school, home, and community [5,7]. In particular, schools are considered a crucial setting to reach and impact adolescents’ health-related behavioral outcomes, motivation, and healthy lifestyles [8,9].

*The School Wellness Integration Targeting Child Health (SWITCH)* is an evidence-based childhood obesity prevention intervention (www.iowaswitch.org), centered on building school capacity to implement comprehensive wellness programming as a means to reach students and parents. The original Switch trial that took place at 10 schools across two states showed efficacy to change 3rd and 4th grade students’ healthy-living behaviors (i.e., physical activity, screen time, fruits and vegetable consumption) [10]. The Switch program was subsequently modified into SWITCH to function in a cost-effective manner, by transitioning to an online platform for data input and behavioral assessment procedures. A formative evaluation of the program demonstrated that the online distribution of SWITCH worked similarly as the traditional print program, for a lower financial cost [11]. Subsequently, with grant support from the United States Department of Agriculture (USDA; grant#: 2015-68001-23242), the SWITCH program underwent major modifications and has been disseminated in dozens of elementary schools in Iowa to promote students’ “Do” (switch up to 60 min or more of physical activity a day), “View” (switch down to 2 h or less of screen time a day), and “Chew” behavioral goals (switch up to 5 or more servings of fruits and vegetables a day). The SWITCH Implementation Process was established to build the capacity of school wellness leaders to lead school wellness programming [12]. As an application of Healthy Youth Places (HYP) Framework [13,14], the SWITCH Implementation Process was designed as a whole-of-school approach to health promotion [15,16,17] that included training and technical support to school SWITCH implementation teams [12]. The school implementation team was a hub for implementation in the classroom, lunchroom and physical education (PE) [12]. Supporting the implementation process was an online health promotion behavioral-tracking platform [18], which has shown good utility to change physical activity outcomes [18,19,20].

Building upon the foundation of the SWITCH intervention for elementary school wellness and child health, the middle school version was recently developed, namely, SWITCH–Middle School (MS). Following the same implementation process framework [12], SWITCH–MS embraced the guiding design, principles, and functionalities (upon making necessary adaptations to fit the structure of middle schools and the developmental appropriateness of adolescents), and was administered in tandem with the elementary school SWITCH for implementation in the 2018‒2019 iteration (IRB. no 18-506). The programming and evaluation of SWITCH in this iteration were guided by an implementation science perspective to simultaneously understand the implementation and effectiveness outcomes [21,22]. The implementation science perspective emphasizes implementation and effectiveness as equally important outcomes, especially in multi-year, large-scale complex interventions such as SWITCH [23,24,25]. The latest SWITCH elementary school intervention demonstrated modest impact of implementation on capacity change at the targeted schools [21,22], and this capacity change was not predicated on substantial training and support [21]. However, the implementation and effectiveness outcomes of the newly added SWITCH–MS remain unclear. Thus, the purpose of this study was to evaluate the implementation and effectiveness outcomes of SWITCH–MS in schools of two distinct treatment conditions: inexperienced (no prior exposure to SWITCH) vs. experienced schools (some prior exposure). Testing the relative differences in implementation and effectiveness outcomes between the two conditions will determine whether prior exposure to SWITCH during elementary school years is a significant factor for the dissemination of SWITCH–MS intervention to a broader population of schools.

## 2. Materials and Methods

### 2.1. Design, Settings and Participants

An effectiveness-implementation hybrid type 3 quasi-experimental design was followed to implement and evaluate the SWITCH–MS intervention [24]. A type 3 hybrid design is primarily concerned with determining the utility of an implementation but also has a secondary aim at assessing clinical outcomes to determine effectiveness [24]. For pragmatic reasons, we did not recruit a true control group and randomize the schools to determine effectiveness; therefore, the design was regarded a quasi-experimental design. Figure 1 illustrates the enrollment flow of the study. Nine schools were categorized in one of two conditions: inexperienced (no prior exposure to SWITCH) and experienced schools (some prior exposure) [24]. Two schools were excluded from the study due to reasons unrelated to the intervention. Doing an intent-to-treat analysis would be unnecessary because the two schools did not complete the protocol and could not contribute to data at posttest. Both of these two middle schools were located in rural areas with similar socioeconomic status (57% vs. 48% eligible for free or reduced-price lunch program). One school was larger (enrollment: 435 vs. 148 students), which was more racially diverse than the other school (96% vs. 55% White/Caucasian). The seven remaining schools participated in the project to receive SWITCH–MS either as experienced (*n* = 3 with 110 students; 51 boys, 59 girls; 54 in 6th grade, 38 in 7th grade, 18 in 8th grade) or inexperienced schools (*n* = 4 with 267 students; 119 boys, 148 girls; 199 in 6th grade, 37 in 7th grade, 31 in 8th grade). Middle schools that had counterpart elementary schools or fourth and/or fifth grades that participated in the SWITCH elementary school project in the year before (2017–2018) were deemed “experienced” schools, or otherwise as “inexperienced” schools.

### 2.2. SWITCH–MS Intervention Process

The SWITCH–MS intervention program was expanded from the SWITCH elementary school version, an evidence-based intervention [10]. Both interventions are guided by the SWITCH implementation process framework [12] adapted from the HYP framework [13]. The broad objective is to provide a whole-of-school self-sustaining infrastructure that features enhanced school wellness capacity, conducive implementation of best practices and quality elements in schools and homes, and pursuing goals and objectives in each setting [12,17]. The school wellness implementation team was three or more individuals drawn from implementation settings (i.e., PE, health education classroom, lunchroom) and administrators, one of whom assumed the school wellness team leader position. The SWITCH–MS implementation process began with a series of preparatory webinars in summer/fall of 2018 and a one-day in-person training conference occurred at the Iowa State University campus in early November 2018. Three implementation team members were required to attend the conference, where they received training about the SWITCH–MS quality elements, best practices, background and rationale, and key resources. The Iowa State University Outreach and Extension 4-H Youth Development staff (https://www.extension.iastate.edu/4h/) who facilitate county-level 4-H programming also attended the conference to learn about SWITCH–MS and then were tasked to work with the implementation teams from specific schools within their respective counties to set up SWITCH–MS goals and develop action plans. The 4-H stands for head, heart, hands, and health, and it is America’s largest youth development organization that empowers nearly six million young children with the skills to lead for a life time (https://4-h.org/about/what-is-4-h/). The Iowa State University Outreach and Extension 4-H was a partner of the SWITCH–MS project, who contributed to recruitment, training, and engaging of schools. They were considered key agents of change throughout the implementation of SWITCH–MS. A recent study has shown that the involvement of 4-H extension staff is important to change school capacity for wellness programing [26]. Throughout the process, implementation targeted health education, PE, and lunchroom change as supported by curriculum modules and a robust website that facilitates access to resources and students’ goal-setting and behavioral tracking (www.iowaswitch.org). The schools completed baseline assessments before the spring 2019 semester started and then launched the 12-week SWITCH–MS programming. The intervention spanned from late January to early May (including interruptions of spring break and inclement weather). The schools completed post-test data collection in May 2019.

### 2.3. Implementation and Effectiveness Outcomes

#### 2.3.1. Whole-of-School Implementation Outcomes

The whole-of-school implementation outcomes of the SWITCH–MS were self-reported by the key informants using a 21-item survey administered toward the end of implementation (week 12). Specifically, 9 questions assessed the implementation of best practices by the school team (3 items for each setting, including lunchroom, health education, and PE) and 12 questions assessed awareness of SWITCH–MS goals and staff engagement level (6 questions for each construct). These questions were answered on a 3-point scale. For example, a question that measures engagement in PE classes is phrased as: “How would your team rate the overall degree of engagement in PE?” Possible choices were: “1 = Low (little to no involvement)”, “2 = Moderate (some involvement but not enough to be a key role player in SWITCH)”, and “3 = High (Lots of involvement and playing a key role in SWITCH implementation)”.

#### 2.3.2. Classroom Implementation Outcomes

In health education classrooms, students were asked to track their “Do” (at least 60 min per day), “View” (no greater than 2 h of non-educational screen time per day), and “Chew” behaviors (5 or more servings of fruits and/or vegetables) each week, using the sliding bar on the project website (www.iowaswitch.org), to inform the classroom-based implementation outcome. Records of completed vs. not completed trackers were saved to the website. Completion rate among students per class was calculated to show the level of tracking behavior. The class-level tracking rate was aggregated by school and then by condition (inexperienced vs. experienced) for subsequent analysis. As documented in prior research, students’ behavioral tracking rate has been shown to be an important element of SWITCH implementation and functionality [11,18,22].

#### 2.3.3. Effectiveness Healthy-Living Behavior Outcomes

Three healthy-living behaviors including “Do” (PA), “View” (screen-based sedentary behaviors), and “Chew” (fruits and vegetable consumption) were measured using the validated Youth Activity Profile (YAP) to determine the effectiveness of SWITCH–MS. YAP has 20 questions with 10 capturing “Do” (5 on PA at school and 5 on PA out of school) and 5 each capturing “View” and “Chew” behaviors. The questions were answered mainly on a 5-point scale. For example, a question that measures “Do” at school is phrased as: “How many days did you walk or bike to school? (if you can’t remember, try to estimate”. Choices were: “A. 0 days (never)”; “B. 1 day”; “C. 2 days”; “D. 3 days”; “E. 4‒5 days (most every day)”. Previous research reported that the 15 questions of YAP assessing PA and sedentary behavior could produce relatively accurate group-level estimates of PA and sedentary in comparison to objective measures [27,28]. For example, one recent study reported that upper and lower limit of 95% CI for YAP predicted scores were within 10–20% of accelerometer-estimated scores [28]. The 5 questions on the “Chew” behavior were added upon the original validation of YAP, and showed small to moderate correlations (*r* = 0.27) with a validated instrument for concurrent validity [29].

### 2.4. Data Collection

Students’ healthy-living behaviors (i.e., “Do”, “View”, “Chew”) were assessed using YAP before and after SWITCH–MS implementation. Students completed the questionnaire online at their school’s media center, or in their gymnasium, using laptop computers under the supervision of their teachers. The teachers received training through preparatory webinars on how to administer the questionnaire. Students were encouraged to ask questions when needed, and then completed the questionnaire independently. In addition, students were prompted by their health education classroom teachers to complete the “Do”, “View”, and “Chew” behavioral trackers each week during the 12 weeks of SWITCH–MS implementation. For the staff survey, the core school implementation team staff or key informants were asked to complete the survey online toward the end of the SWITCH–MS implementation (week 12). All data were automatically recorded and saved in the online server and then downloaded during summer 2019 for cleaning and processing.

### 2.5. Data Analysis

Descriptive statistics (M, SD, N, percentage) were obtained to evaluate the implementation of SWITCH–MS. First, for whole-of-school implementation evaluation, descriptive statistics for staff-reported implementation of best practices (including PE, health, and lunchroom), awareness, and engagement were compared between inexperienced and experienced schools. For classroom implementation evaluation, a line chart was drawn to illustrate the weekly tracker completion rate for inexperienced vs. experienced schools across the 12 weeks. To determine the effectiveness of SWITCH–MS, a series of three-level linear mixed models (i.e., time nested within student nested within schools) were specified, where student-level healthy-living behaviors were entered as dependent variables (i.e., “Do” at school, “Do” out of school, “View”, and “Chew” behaviors, respectively), with one dependent variable per model. Specifically, in model 1, we tested the null or no predictor model with randomly varying intercept at student and school–level. In model 2, we tested the fixed effects of time (pre vs. posttest), condition (inexperienced vs. experienced schools), and time–by–condition interaction with randomly varying intercepts at student and school levels. In model 3, we added gender and grade as student-level covariates to the model 2. In model 4, we added one of the four school–level covariates (i.e., behavioral tracking, staff–reported best practice implementation, awareness, and engagement, respectively) to model 3. Socioeconomic status was not entered as a school level predictor due to its small variation between schools (see FARM% statistics in Table 1). Least squared means and standard errors for time–by–condition interaction were obtained from model 4 where behavioral tracking was entered as a school-level predictor. Line charts were drawn to depict the time–by–condition interaction effects for “Do” at school, “Do” out of school, “View”, and “Chew” behaviors. [17]. Unstandardized regression coefficient estimate (B), standard error (s.e.), *p* value, and intra-class correlational (ICC) coefficient were reported as results from model 4 for each of the healthy-living behaviors. The sequential three-level linear mixed modeling described above was performed by following Heck et al.’s tutorials [30]. SPSS 26.0 (IBM: Armonk, NY, USA) was used for data analyses and Microsoft Excel 2016 was used to draw line charts. Alpha was set at the 0.05 level for significance testing.

## 3. Results

### 3.1. Implementation Outcomes

Table 2 shows the staff-reported implementation levels of SWITCH–MS best practices (including health education, PE, and lunchroom curricular modules), awareness of SWITCH goals, and engagement in accomplishing these goals for inexperienced vs. experienced schools. The average scores for these implementation indicators were commensurate between the two groups of schools, although experienced schools showed higher levels of engagement and inexperienced schools showed higher implementation in the lunchroom.

Figure 2 illustrates the weekly tracker completion rate over the 12 weeks of SWITCH–MS implementation at inexperienced vs. experienced schools. Overall, both groups of schools showed low to moderate tracking rates (below 50%). Experienced schools started at a moderate level of behavioral tracking rate (47.5%) and sustained at the range of 31.3‒44.3% between weeks 3 and 12. Inexperienced schools’ behavioral tracking rate started low (11.7% in week 1 and 6.6% in week 2) but gradually increased from week 3 and sustained at a level that was comparable to the experienced schools’. As shown in Figure 2 (see standard errors), variation of behavioral tracking rate between schools per group was relatively large

### 3.2. Effectiveness Outcomes

Four sequential linear mixed models were conducted for each of the behavior outcome variables (i.e., “Do” at school, “Do” out of school, “View”, and “Chew” behaviors). Table 3 show the results from the final model (i.e., model 4) for each of the four healthy-living behaviors. Model 1 (i.e., the null/no predictor model) across the four behavioral outcomes showed small clustering effect at the school-level which accounted for 2.0‒9.8% total variances, so student- and school-level predictors were subsequently added for further testing. Model 2 (time, condition, and time-by-condition interaction) showed a significant time effect for “Do” at school (*p* = 0.01), “Do” out of school (*p* < 0.01), and “View” behaviors, favoring posttest behaviors (*p* = 0.02). These models showed a small clustering effect, with 1.8‒9.6% variances accounted for at the school level. Model 3 added gender and grade as student-level predictors to model 2, and model 4 added behavioral tracking, staff-reported implementation of best practice, awareness, and engagement, respectively. Compared to girls, boys showed significantly higher “Do” behaviors (at school *p* < 0.01, and out of school *p* = 0.03) but also higher “View” behavior (*p* < 0.01). By grade, 6th grade students displayed significantly higher “Do” behaviors, higher “Chew” behaviors, and lower “View” behaviors (*p* < 0.01) than 8th grade students; while 7th grade students also showed higher “Chew” behaviors than 8th grade students (*p* = 0.04). Behavior tracking, as a school-level covariate, only seemed to matter to “Do” out of school behavior (*p* = 0.03). To save space, Table 3 only reports behavioral tracking rate as the only school-level predictor as these school-level predictors showed similar results. School-level clustering ranged 3.8‒17.3% for model 3 and 0.7‒20.8% for model 4, across the four behavioral outcomes.

Figure 3 illustrates the descriptive results for “Do” (at school and out of school), “View”, and “Chew” behaviors by time (pretest vs. posttest) and condition (inexperienced vs. experienced schools) using the least square means and standard errors obtained from model 4 of four separate linear mixed models (adjusted for student- and school-level covariates). Overall, both inexperienced and experienced conditions showed favorable temporal changes for “Do” at school, “Do” out of school, and “View” behaviors (lower values indicate lower screen-based sedentary behaviors).

## 4. Discussion

The purpose of this study was to evaluate the implementation and effectiveness of the SWITCH–MS intervention, as a whole-of-school intervention. The implementation and effectiveness outcomes were compared between inexperienced vs. experienced schools, which showed similarities and differences that are theoretically and practically meaningful. The findings and their implications are discussed below.

The implementation evaluation of SWITCH–MS revealed some interesting findings about the implementation outcomes. First, the whole-of-school implementation of SWITCH–MS components were similar and commensurate between the two conditions of schools for utilizing best practices in PE, health classroom, and lunchroom; being cognizant of the SWITCH–MS goals at multiple levels (from teachers and lunchroom staff to parents’ involvement); and staying engaged in achieving these goals (see Table 2). However, on average, the experienced schools (*n* = 3) showed slightly higher levels of engagement, but lower levels of best practice implementation in the lunchroom than the inexperienced schools (*n* = 3). Inferential statistics were not conducted on these implementation outcomes due to small sample size at the school level (*n* = 6). Furthermore, classroom-based implementation—as represented by weekly behavioral tracking rate for “Do”, “View”, and “Chew” behaviors—was better in the experienced schools during the first two weeks of implementation than that in the inexperienced schools, but the group differences were attenuated between weeks 3 and 12. As previously mentioned, behavioral tracking is an essential element of the SWITCH programing [12,18]. Previous evaluation of the SWITCH elementary school intervention showed staff engagement was a key factor of boosting and sustaining students’ behavioral tracking rate [12]. The present implementation outcomes for SWITCH–MS are somewhat consistent with the findings from the 2017‒2018 iteration of the SWITCH elementary school intervention which showed that SWITCH was implemented to a similar extent between the standard and enhanced conditions [21,22], showing SWITCH–MS can be easily implemented without the necessity of prior exposure or additional training/support. This finding is encouraging to the future dissemination and implementation of SWITCH–MS in broader schools.

The effectiveness evaluation demonstrated that SWITCH–MS intervention was associated with significant temporal changes in most of the healthy-living behaviors (“Do” at school, “Do” out of school, and “View” behaviors) among middle school students at both inexperienced and experienced schools. Our three-level linear mixed models adjusted for student- and school-level predictors in order to accurately determine the intervention effect. The non-significant time-by-condition interaction effects for the healthy-living behaviors suggest that prior exposure to SWITCH in upper elementary school years did not matter when it comes to the effectiveness of SWITCH–MS on these behaviors. In other words, SWITCH–MS can render similar influence on promoting middle school students’ healthy-living behaviors (“Do” and “View”), which is not predicated on prior experiences. These results are consistent with the previous findings shown in the original elementary school Switch efficacy trial, verifying the potential of SWITCH programing in influencing school wellness and child health [10,12,18].

Taken together the implementation and effectiveness evaluation, this study provides evidence of SWITCH–MS, as an adolescent obesity prevention intervention. The findings verify the importance of following the SWITCH Implementation Process framework for implementation across environmental settings, attending preparatory webinars and in-person training, as well as the involvement of 4-H staff engagement [12,26]. Making adaptations to fit the needs and structures of middle schools and the developmental levels of adolescents was also a critical step, however, to provide a more appropriate implementation process for the SWITCH–MS intervention. The findings from this study corroborate that schools can be an effective setting to promote and shape adolescents’ healthy-living behaviors [8,9]. Conducive environmental affordances and wellness policies, which typically are lacking or inadequate and require intentional promotion in most schools, are needed to enhance implementation and effectiveness outcomes for multi-component, ecological interventions such as SWITCH–MS.

Despite the demonstrated implementation and effectiveness of the SWITCH–MS intervention, we acknowledge several limitations of this study. First, in this effectiveness-implementation hybrid type 3 design quasi–experimental study [24], we did not recruit schools to serve as true controls, which limits our ability to determine the true effectiveness of SWITCH–MS in changing healthy-living behaviors. As a result, any temporal change observed in this study could be due to seasonality effects. However, this research design was pragmatic in public health research, as Switch at the elementary school level previously established efficacy [10] and the SWITCH–MS is a derivative of its elementary predecessor. Therefore, the observed temporal changes in healthy-living behaviors between the two conditions can be viewed as relative effectiveness. Another noticeable limitation of this study is the small sample size and heterogeneity of enrollment size at the school level. Due to substantial missing data at two schools, we were only able to utilize data collected from seven schools. We wish we recruited more schools across geographical regions but only had limited resources to conduct this pilot study. In addition, this study used self-reported measures to assess the variables. We acknowledge self-report measures may not be as accurate as objective measures. However, in a large-scaled intervention study like this one, collection of self-reported data is more feasible.

## 5. Conclusions

This evaluation study has provided evidence for the implementation and effectiveness outcomes of SWITCH–MS, as a whole-of-school intervention, in changing middle school wellness and adolescents’ healthy-living behaviors. For implementation outcomes, the two groups of schools showed similar levels of self-reported best practices, awareness, and engagement. Classroom-based implementation, as represented by the behavioral-tracking practice, was also similar between the two groups of schools after the intervention was launched beyond two weeks. For effectiveness outcomes, students’ physical activity levels both during and out of school, and screen-time-based sedentary behaviors improved over the 12 weeks of intervention, regardless of treatment groups. Based on the evidence, we conclude that the implementation and effectiveness of SWITCH–MS are not predicated on students’ prior experience, although some implementation indicators (e.g., behavioral tracking) may favor the experienced schools than inexperienced schools. The findings shed light on future implementation and dissemination of the SWITCH–MS intervention in broader schools.

## Figures and Tables

**Figure 1 ijerph-17-05401-f001:**
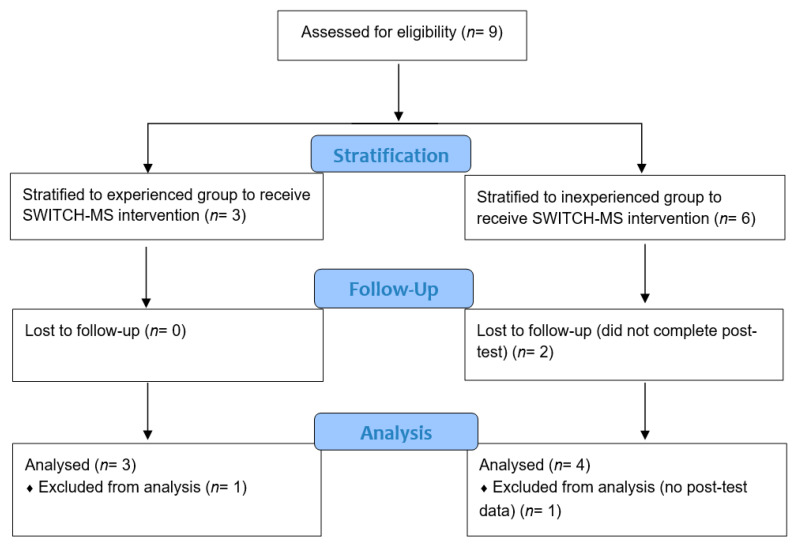
CONSORT Flow Chart for Enrollment. Note. The study followed the effectiveness-implementation hybrid type 3 quasi-experimental design.

**Figure 2 ijerph-17-05401-f002:**
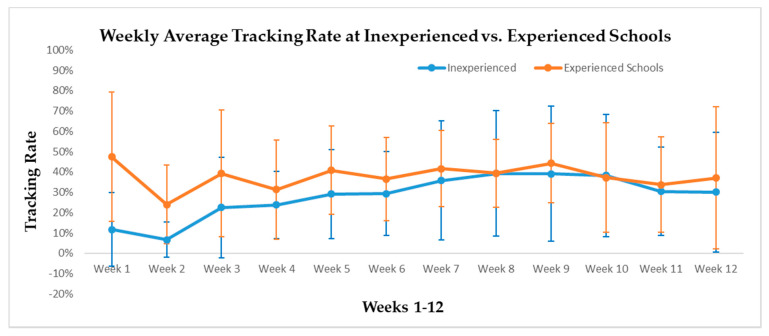
Behavior tracking rate across the 12 Weeks for Inexperienced (*n* = 3) and Experienced (*n* = 3) Schools.

**Figure 3 ijerph-17-05401-f003:**
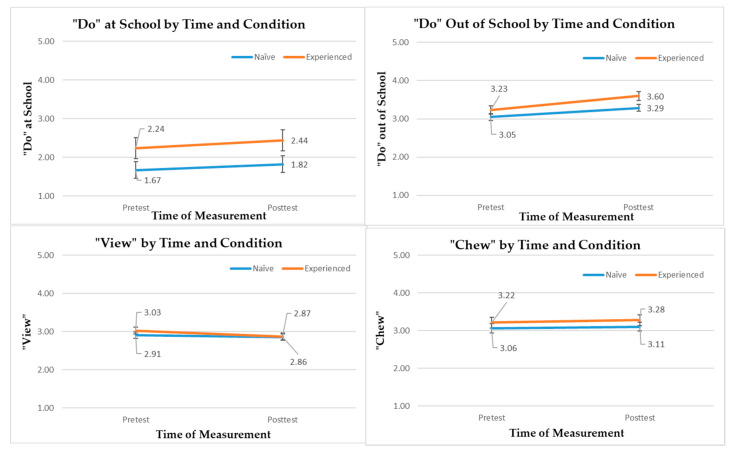
“Do”, “View” and “Chew” Behavioral Outcomes by Time and Group. Error bars denote standard errors. Note, lower values indicate lower level of screen-based sedentary behaviors for “View” behavior.

**Table 1 ijerph-17-05401-t001:** Characteristics of the Inexperienced and Experienced Schools.

School #	Group	Locale	Enrollment	%White	%Male	%FARM
1	Inexperienced	Rural: Distant	227	95%	54%	21%
2	Inexperienced	Rural: Distant	97	99%	46%	22%
3	Inexperienced	City: Mid-size	737	30%	49%	74%
4	Inexperienced	Rural: Distant	189	93%	49%	28%
5 *	Experienced	Town: Fringe	30	92%	45%	---
6	Experienced	Rural: Distant	270	89%	54%	34%
7	Experienced	Rural: Remote	70	96%	54%	26%

**Note**. #: school number; %FARM: percentage of student eligible for free or price-reduced meals. * School 5 is a private school so no %FARM was calculated. Data were retrieved from www.nces.ed.gov.

**Table 2 ijerph-17-05401-t002:** Whole-of-School Implementation Outcomes for Best Practices, Awareness, and Engagement (M ± SD).

Group	Implementation of the Best Practices			Awareness	Engagement
	Health	PE	Lunchroom		
Inexperienced (*n* = 3)	1.9 ± 0.4	1.4 ± 0.2	2.2 ± 1.1	2.2 ± 0.2	1.9 ± 0.3
Experienced (*n* = 3)	1.8 ± 0.4	1.4 ± 0.5	1.7 ± 0.6	2.3 ± 0.3	2.3 ± 0.3

**Note**. One of the four inexperienced schools did not complete the survey. PE: physical education.

**Table 3 ijerph-17-05401-t003:** Results from Four Sequential 3-Level Linear Mixed Models for “Do” at School as Outcome Variable.

Behavior Outcome	Predictors	B	*s.e.*	*df*	*t*	*p*	95% CI		Variances Explained at Levels 1, 2, 3
							Lower	Upper	
“Do” at School	Intercept	2.04	0.41	4.83	4.96	**<0.01**	0.97	3.11	33.5%, 45.7%, 20.8%
	Time (T1)	−0.20	0.07	375.05	−2.83	**<0.01**	−0.34	−0.06	
	Condition (inexp.)	−0.62	0.35	4.32	−1.79	0.14	−1.55	0.31	
	Time×Condition (T1×inexp.)	0.05	0.08	374.93	0.55	0.59	−0.12	0.21	
	Gender (boys)	0.23	0.08	383.42	3.09	**<0.01**	0.08	0.38	
	Grade (6th G)	0.85	0.13	379.46	6.34	**<0.01**	0.59	1.11	
	Grade (7th G)	−0.10	0.14	394.64	−0.73	0.46	−0.37	0.17	
	Tracking	0.14	0.85	4.08	0.17	0.88	−2.20	2.48	
“Do” out of School	Intercept	2.99	0.20	16.25	15.03	**<0.01**	2.57	3.41	45.7%, 53.6%, 0.7%
	Time (T1)	−0.37	0.09	375.08	−4.12	**<0.01**	−0.54	−0.19	
	Condition (inexp.)	−0.31	0.14	5.43	−2.29	0.07	−0.65	0.03	
	Time×Condition (T1×inexp.)	0.13	0.11	374.98	1.21	0.23	−0.08	0.34	
	Gender (boys)	0.20	0.09	384.25	2.17	**0.03**	0.02	0.37	
	Grade (6th G)	0.60	0.15	148.25	3.98	**<0.01**	0.30	0.89	
	Grade (7th G)	0.23	0.16	379.02	1.44	0.15	−0.09	0.55	
	Tracking	0.94	0.30	4.31	3.09	**0.03**	0.12	1.76	
“View” Behavior	Intercept	3.14	0.16	9.89	19.48	**<0.01**	2.78	3.50	46.5%, 52.4%, 1.1%
	Time (T1)	0.16	0.07	374.98	2.25	**0.03**	0.02	0.30	
	Condition (inexp.)	−0.01	0.11	3.74	−0.08	0.94	−0.33	0.31	
	Time×Condition (T1×inexp.)	−0.11	0.08	374.88	−1.35	0.18	−0.28	0.05	
	Gender (boys)	0.21	0.07	384.16	2.98	**<0.01**	0.07	0.35	
	Grade (6th G)	−0.46	0.12	139.81	−3.90	**<0.01**	−0.69	−0.23	
	Grade (7th G)	−0.22	0.13	379.96	−1.73	0.08	−0.47	0.03	
	Tracking	−0.60	0.25	2.96	−2.35	0.10	−1.42	0.22	
“Chew” Behavior	Intercept	2.86	0.23	6.75	12.33	**<0.01**	2.31	3.42	31.7%, 61.9%, 6.5%
	Time (T1)	−0.05	0.06	375.30	−0.90	0.37	−0.17	0.06	
	Condition (inexp.)	−0.16	0.18	4.61	−0.90	0.41	−0.65	0.32	
	Time×Condition (T1×inexp.)	0.01	0.07	375.15	0.11	0.91	−0.13	0.14	
	Gender (boys)	−0.10	0.07	390.33	−1.44	0.15	−0.24	0.04	
	Grade (6th G)	0.64	0.12	333.44	5.17	**<0.01**	0.40	0.89	
	Grade (7th G)	0.26	0.13	405.07	2.02	**0.04**	0.01	0.51	
	Tracking	0.65	0.44	4.09	1.48	0.21	−0.56	1.87	

**Note**. T1 = Time 1 or pretest; inexp. = inexperienced schools condition. Levels 1, 2, 3 correspond to time-, student-, and school-levels. In these models where predictors were categorical variables (time, condition, gender, and grade), time 2 (or posttest), experienced condition, girls, 8th grade was used as referent group. Bold values denote statistically significant or *p* < 0.05

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
