# Peer review of "Evaluating the Implementation and Effectiveness of the SWITCH–MS: An Ecological, Multi-Component Adolescent Obesity Prevention Intervention"

_ijerph, 2020, doi:10.3390/ijerph17155401_

Round 1

Reviewer 1 Report

Overweight and obesity in adolescent are a major public health issue. To build an efficient and simple educational program able to lead to lasting modifications for a healthier lifestyle is a crucial challenge. So, the paper could be of interest.

But a lot of weaknesses can be found reading the paper which remain unaddressed in the methodology and/or discussion chapters.

Presentation can be improved with figures (1 and 2) showing black and white differences as it was impossible to identify the two compared populations on the black and white printing of the downloaded pdf document.

The "4H- extension and programming" have to be made more explicit.

The methodology of the trial (“implementation-effectiveness hybrid type 3 quasi experimental design”) has to be described in the chapter methodology and clearly discussed in the chapter discussion.

The results are quite poor and appear to be limited to some levels of the educational process; such levels remain unclear for non-US persons and have to be address for such readers.

A major weakness could be the heterogeneity, or irrelevance of the comparisons made. Comparing schools 1 and 2 on one hand to schools 6 and 7 on the other hand could probably be more relevant as the situations and sizes are quite similar for such groups. The introduction of a very small private school (high economical level) and of a big public school (with a high level of poverty and major ethnic differences) make probably the results confusive.

It would have been good to register this study in “ClinicalTrial.gov” and to get the approval of an independent ethical committee.

Reviewer 2 Report

This is an interesting and important study concerning the obesity epidemics, and although the design was not optimal (e.g., small sample size, lack of control group, small geographical variation) to provide generalisation, there are merits in conducting this challenging research on a relatively "large scale" and ecological levels. I only have minor comments/suggestions.

Introduction

line 69: Can the authors explain the meaning of “success”? Previous sentences do not inform about the effectiveness of the project.

Methods

line 95: Can the authors provide details about the 2 schools that were excluded, including the reasons for having missing data? Shouldn’t this be part of the analysis?

I suggest a flow diagram of the study (e.g., participants, design) so that readers can follow it easily.

line 157: Please provide the statistics so that readers can appraise “the sound concurrent validity” statement.

Results

Sub-heading “Effectiveness” is missing.

Discussion

Limitations should include also the small sample size and the relatively small geographic variation of the implemented program that limits generalization (along with the absence of control group as already acknowledge by the authors).

Conclusion

This is somewhat vague. Can the authors provide a summary of further objective findings other than the results of previous exposure of schools to the SWITCH program?

Reviewer 3 Report

I have carefully reviewed the manuscript entitled "Evaluating the Implementation and Effectiveness of 3 the SWITCH–MS: An Ecological, Multi-Component 4 Adolescent Obesity Prevention Intervention”. This quasi-experimental study aimed to evaluate the implementation and effectiveness of an ecological, multi-component adolescent obesity prevention intervention called SWITCH – MS. The authors went to great lengths to design and implement a 12-week intervention. The design and methodology were appropriate to conduct this intervention. In my opinion, the paper can be of great value to IJERPH and readers as well in the current version. I have only very two minor comments and some references that authors should use to improve the state of the art in the introduction and to discuss better some results of their study.

Introduction:

- Overall, the introduction and the paper is very well written. The authors use adequate and recent bibliography on the study topic. In addition, they make a brief review of the state of art on ecological and multi-component school-based intervention in children and adolescents. In this sense, I would like to recommend some similar interventions based on the ecological model, multilevel and multi-component that have been carried out recently in European students. I thought that these studies, which have been published in high-quality JCR journals in the last years may improve the state of the art of your introduction, especially, in the first paragraph of the introduction. Likewise, they can be useful to improve the discussion section. These suggested references to be added are:

Sevil-Serrano, J., Aibar, A., Abós, Á., Generelo, E., & García-González, L. (2020). Improving motivation for physical activity and physical education through a school-based intervention. The Journal of Experimental Education, 1–21. https://doi.org/10.1080/00220973.2020.1764466

Sevil, J., García-González, L., Abós, Á., Generelo, E., & Aibar, A. (2019). Can high schools be an effective setting to promote healthy lifestyles? effects of a multiple behavior change intervention in adolescents. Journal of Adolescent Health, 64(4), 478–486. https://doi.org/10.1016/J.JADOHEALTH.2018.09.027

Methods:

Participants

- I would like that authors added more information about gender, age, and socioeconomic level of participants.

Results

- Please Table 3 should be shown in horizontal layout to a better understanding and reading.

Congratulations for this well-conducted and important paper!

Round 2

Reviewer 1 Report

Manuscript acceptably improved accordons reviewer’s observations.